# The Use of Ozone as an Eco-Friendly Strategy against Microbial Biofilm in Dairy Manufacturing Plants: A Review

**DOI:** 10.3390/microorganisms10010162

**Published:** 2022-01-13

**Authors:** Felice Panebianco, Selene Rubiola, Pierluigi Aldo Di Ciccio

**Affiliations:** Department of Veterinary Sciences, University of Turin, Largo Braccini 2, Grugliasco, 10095 Torino, Italy; felice.panebianco@unito.it (F.P.); selene.rubiola@unito.it (S.R.)

**Keywords:** microbial biofilm, dairy plants, dairy products, ozone, innovative technologies, foodborne pathogens, spoilage bacteria, food quality, food safety

## Abstract

Managing spoilage and pathogenic bacteria contaminations represents a major challenge for the food industry, especially for the dairy sector. Biofilms formed by these microorganisms in food processing environment continue to pose concerns to food manufacturers as they may impact both the safety and quality of processed foods. Bacteria inside biofilm can survive in harsh environmental conditions and represent a source of repeated food contamination in dairy manufacturing plants. Among the novel approaches proposed to control biofilm in food processing plants, the ozone treatment, in aqueous or gaseous form, may represent one of the most promising techniques due to its antimicrobial action and low environmental impact. The antimicrobial effectiveness of ozone has been well documented on a wide variety of microorganisms in planktonic forms, whereas little data on the efficacy of ozone treatment against microbial biofilms are available. In addition, ozone is recognized as an eco-friendly technology since it does not leave harmful residuals in food products or on contact surfaces. Thus, this review intends to present an overview of the current state of knowledge on the possible use of ozone as an antimicrobial agent against the most common spoilage and pathogenic microorganisms, usually organized in biofilm, in dairy manufacturing plants.

## 1. Introduction

The management of microbial contamination in the food chain is the main goal of the modern food industry. In particular, the control of contamination by spoilage bacteria, such as *Pseudomonas* spp., as well as pathogenic bacteria, such as *Listeria monocytogenes*, *Salmonella* spp., etc., represents a big challenge for the dairy industry. If environmental conditions are suitable, these spoilage and pathogenic bacteria can quickly form biofilm and persist in dairy processing plants. Microorganisms in a biofilm state, indeed, can survive in harsh environmental conditions [1,2,3]. In this context, the presence of biofilms that include spoilage and pathogenic bacteria constitutes a source of repeated food contamination with consequent issues related to the shelf life and safety of dairy products [4]. Microbial biofilms can be found wherever moisture and sufficient nutrients are available. In this regard, food processing plants, especially dairy plants, become an ideal environment for biofilm formation by several microorganisms [5]. It is well known that the presence and control of microbial biofilm represent a big issue for the whole dairy food chain, from milk collection to waste management [6]. Maintaining adequate hygienic conditions is crucial for preventing biofilm formation by pathogenic and/or spoilage microorganisms in food environments. Currently, biofilm development in food processing facilities, including dairy plants, is controlled by using common biocides, such as quaternary ammonium compounds, alcohols, aldehydes, peracetic acids or chlorine compounds, and by following daily accurate sanitation procedures. Anyway, bacteria inside biofilm are much more resistant to antimicrobials. The biofilm, in fact, can protect bacteria against the action of cleaning agents and disinfectants [4,7]. However, the application of biocide compounds has some drawbacks, including detrimental impacts on the environment and deleterious consequences for human health [8]. In addition, the use or misuse of biocides could presumably act as selection pressures for increased microbial resistance to antimicrobial compounds, and the sanitizers may lose effectiveness over time. This phenomenon, known as antimicrobial cross-resistance [3,9,10,11,12], is considered as one of the most relevant risks to public health worldwide [13].

As a result, the interest in additional or alternative compounds has increased in the last few years. Considering the concerns for the microbial contamination and biofilms formed by spoilage or pathogenic bacteria and the high cost of managing these issues in processing plants, the identification of novel strategies represents one of the most critical challenges that the modern food industry will face in the following years. Therefore, several innovative strategies, such as the use of antimicrobial peptides, bacteriophages, bacteriocins, essential oils, hydrostatic pressure, cold plasma, hot steam, and ultrasonication, have been proposed to prevent and control microbial contamination in food environments [1,14,15,16,17]. Among these new methods, ozone (O_3_) seems to be a promising tool to prevent microbial contamination by spoilage or pathogenic bacteria. Ozone, in fact, is characterized by high antimicrobial activity and it is recognized as an eco-friendly technology due to its low environmental impact [8,18,19,20,21,22]. In addition, ozone breaks down into oxygen without leaving dangerous residues on food or food-contact surfaces [20,23,24]. Ozone, when used in a gaseous state, can reach niches and other “dead zones” within food processing environments, where microorganisms can persist more easily after conventional cleaning and disinfecting programs [25,26]. Recently, several studies have highlighted that ozone may effectively control microbial contamination and biofilms in the food industry and can be a valid alternative or an additional tool to conventional strategies based on chemical disinfection [19,23,27,28,29,30]. Hence, this review intends to provide an overview of the current state of knowledge on the possible use of ozone as an antimicrobial agent against the most common spoilage and pathogenic microorganisms, usually organized in biofilm, in dairy manufacturing plants.

## 2. Biofilm Occurrence in the Dairy Industry

In a recent review [6], the role of microbial biofilm in the dairy industry has been described extensively. It is well known that bacteria can form biofilm at each point of the production process and in different parts of the processing equipment, including tanks, silos, pipelines, tubes, membranes, walls, plate heat exchangers, etc. [31]. Biofilms of spoilage microorganisms or pathogenic bacteria in milk processing plants are deemed as a major problem. Teh et al. [32], in fact, demonstrated that biofilms formed by *Bacillus licheniformis*, *Streptococcus uberis*, *Pseudomonas fluorescens*, *Pseudomonas fragi* and *Serratia liquefaciens* on the internal surfaces of raw milk tankers may be sources of proteolytic enzymes. The persistence of these microorganisms, especially *Pseudomonas* spp., and the production of thermostable proteases can lead to the potential spoilage of milk during the subsequent steps in the dairy supply chain [32,33]. A review addressed in detail the role of bacterial biofilms as an emerging source of spoilage enzymes in the dairy environment, emphasizing the importance of biofilm control in dairy plants to improve product quality and avoid economic losses [34]. Additionally, the presence of biofilms of pathogenic bacteria in milking equipment and milk storage tanks may increase the risk of causing foodborne illness. In regard to this, Latorre et al. [35] demonstrated that biofilms in milking equipment on a dairy farm can represent a source of *Listeria monocytogenes* contamination in bulk tank milk. Another study showed that biofilm formed by *Staphylococcus aureus* could be an important contamination source for bulk tank milk [36]. Consolidated biofilms were identified as a cause of the persistent contamination of *Campylobacter jejuni* in raw milk [37]. Bacterial biofilms can be found also in membranes used for filtration of milk and other fluids in dairy industries, such as whey and water. This topic has been addressed in detail in a review produced by Anand et al. [38], highlighting how in biofilms detected in this equipment both spoilage and pathogenic microorganisms could coexist for a long time. In the biofilm formed on microfiltration and reverse osmosis membranes, seventeen different bacterial groups were detected, with Proteobacteria identified as the major represented group [39]. Recently, a study carried out by Chamberland et al. [40] clarified the composition of biofilms found in spiral-wound membranes, commonly used in milk processing industries. Proteobacteria, Firmicutes, and Actinobacteria were the three major represented phyla. Proteobacteria were dominant (relative abundances of 63.62 and 62.68%) for membranes used for ultrafiltration of pasteurized milk. In those samples, the highest proportion of Actinobacteria was also detected. *Methylobacterium* spp. was the most abundant genus, while the prevalent Bacilli genera were *Streptococcus* spp., *Enterococcus* spp., and *Lactococcus* spp. *Acinetobacter* spp., *Cronobacter* spp., and *Klebsiella* spp., instead, were the dominant γ-Proteobacteria found. Regarding the spiral-wound membranes used for cheese whey, whey permeate, and water blend filtration, Firmicutes was the most represented phylum, due to the presence of a high Bacilli class ratio. Biofilm formation in heat exchangers may have strong negative effects, as reported by Marchand et al. [31]. The flow of milk in the exchangers, in fact, results in a denaturation of proteins with consequent fouling. This can accelerate the adhesion of microorganisms on the exchangers surface and lead to the formation of biofilms [41]. Sharma and Anand [42] evaluated the biofilms of pasteurization lines in commercial and experimental dairy plants and found the *Bacillus* genus to be predominant in both cases. The same authors emphasized that the assessment of biofilms and the development of efficient sanitization protocols should be part of the HACCP plan in dairy industries. Biofilms can be present also in filler nozzles. In this regard, Mugadza et al. demonstrated that biofilms formed by *Bacillus cereus* in filler nozzles could be the main cause of extended-shelf-life milk contamination [43]. Weber et al. [44] analyzed the biofilms of milking machines of two dairy farms. The outcomes demonstrated high bacterial diversity, including the phyla Actinobacteria, Bacteroidetes, Firmicutes, and Proteobacteria. Biofilms were also detected on several materials, such as stainless steel and plastic, which are used in milking machines and in processing plants.

Microbial biofilms are commonly found in cheese processing environments. In a work conducted by Lee et al. [45], persistent biofilm forming *L. monocytogenes* strains were isolated at different points in Brazilian cheese processing plants, including the cooling chamber (*n* = 16), floor of pasteurization room (*n* = 8), floor of cooling chamber (*n* = 32), plastic crates (*n* = 8), platform of cooling chamber (*n* = 7), surfaces of worker’s gloves (*n* = 3), and brine (*n* = 5). Persistent *L. monocytogenes* strains, organized in biofilm, were detected in Gorgonzola cheese processing plants located in Italy [46]. Bacterial biofilms are commonly detected on wooden shelves used during the ripening process of several traditional cheeses. In this case, biofilms may also have a positive impact on the final products, as these can be composed by the desired dairy lactic acid bacteria. Lactic acid bacteria, in fact, can inhibit the growth of undesirable microorganisms through the production of several antimicrobial compounds [47,48]. A study conducted by Didienne et al. [49] characterized biofilms of ‘gerles’ (wooden vats used in PDO Salers cheese making) and identified their role in milk inoculation and in preventing pathogen development. Biofilms were mainly composed of different lactic acid bacteria, Gram-positive catalase positive bacteria and yeast, while they were not contaminated by *Salmonella*, *L. monocytogenes* or *S. aureus*. These authors showed that wooden surfaces used during the production of these traditional cheeses are a safe system and a microbiologically active tool. Another study demonstrated the positive influence of spontaneous biofilms grown on wooden surfaces on the microbiological, chemical, physical, and sensory characteristics of PDO Vastedda della valle del Belìce cheese [50].

## 3. Ozone

Despite the regular application of cleaning and disinfection plans, bacterial biofilms containing spoilage or pathogenic bacteria are commonly found in dairy manufacturing plants. The biofilm, in fact, protects bacteria against the effects of cleaning agents and disinfectants. Among the innovative strategies, ozone, in its aqueous or gaseous form, is considered to be a promising eco-friendly technology and may be applied as an additional tool to control microbial biofilm in dairy processing environments.

### 3.1. Chemical, Physical Properties, and Antimicrobial Action

Ozone is a blue gas with a characteristic pungent smell and it is known as the second most powerful oxidizing agent after fluorine. An ozone molecule is formed by three oxygen atoms with a central nucleus attached to two equidistant atoms. This structure and the arrangement of the unpaired electrons are responsible for its strong reactivity [27,28]. The high oxidative potential, the instability and the reactivity determine the antimicrobial activity of ozone. Ozone exerts its action on microorganisms attacking the constituents of cell membranes, cell envelopes, cytoplasm, spore coats, and virus capsids [29]. Two principal mechanisms of microorganisms’ destruction by ozone were identified [29]. The first one is related to the oxidative action of ozone on the sulfhydryl groups and amino acids of enzymes, peptides and proteins that lead to the formation of shorter peptides, while the second mechanism involves the ability of ozone to oxidize polyunsaturated fatty acids in peroxides. In this regard, the double bonds of unsaturated lipids of the cell envelope are especially sensitive to ozone action. The degradation of these components causes the destruction of cells and the loss of intracellular contents [27]. After the breakdown of the bacterial cell wall, the polyunsaturated fatty acids that constituted the phospholipids of the cytoplasmic membrane undergo a peroxidation process due to ozone action. The peroxides resulting from this process cause substantial changes to the physical properties of the cell membranes, with subsequent depolarization and inhibition of enzymes and transport proteins [29]. As mentioned above, the high oxidative potential of ozone can also result in the degradation of amino acids, proteins, and nucleic acids. This mechanism comes into action when the first one (destruction of the membrane) fails to cause the cell death, but the destruction of membrane barriers is deemed as the main factor leading to nucleic acid damages and cell death [29,51].

### 3.2. Generation

Generation of ozone is relatively easy and is performed “in situ”, since ozone is characterized by high instability and its storage is difficult [20,52]. Several generation methods exist for the practical applications of ozone, even if the electrical and the photochemical (UV) methods are the most used [29]. With the electric corona discharge method, for example, oxygen molecules pass through an electrical field between two electrodes, and they split generating radicals that produce ozone by combining with the oxygen molecule. Ozone may be generated also by ultraviolet radiation, with the passage of oxygen gas molecules in a short-wave UV light high-energy [52]. Other generation techniques include thermal, chemical, electrolytic, and chemonuclear methods [19].

### 3.3. Parameters That Affect the Antimicrobial Performances

The efficiency of ozone is influenced by several factors, among which the pH is one of the most important. It has been demonstrated that ozone is more stable at low pH. Ozone is completely dissolved at pH values lower than 7, while an increase in pH causes spontaneous ozone decomposition, which leads to the production of highly reactive free radicals. At pH 8, almost half of the introduced ozone is decomposed to various intermediate forms and to oxygen [29]. The generated radicals increase the efficiency [19,53]. Temperature can also affect the solubility and the effectiveness of ozone. Ozone, indeed, becomes less stable and soluble when the temperature increases, while also becoming more reactive [19]. Relative humidity (RH) represents another important parameter for the antimicrobial efficiency of ozone. Pascual et al. [20] reported that the bactericidal effect of gaseous ozone reaches an optimum at RH of 90–95%, while no effect was observed at below 50%. Another aspect to consider is represented by the presence of organic matter, which is known to decrease the effectiveness of ozone, pointing out the importance of avoiding the presence of residual organic materials in the environment for the application of this technology [19,26,53].

### 3.4. Aqueous or Gaseous Form?

Ozone may be applied in dairy industries in aqueous and/or in gaseous form. The form to be used must be carefully chosen according to the needs and structural characteristics of processing plants. Generally, ozone in aqueous solutions seems more effective on food-related microorganisms than ozone in the gaseous phase. Indeed, to achieve a significant effect with ozone in gaseous form, longer exposure times and/or higher concentrations are needed [8,52]. However, it has been reported that molecules of ozone in gaseous form are characterized by a longer half-life and diffusion than molecules in an aqueous solution [54]. These characteristics could be exploited to counteract bacterial biofilm in the hard-to-reach areas of food plants [26].

### 3.5. Legislation

Nowadays, the legislation on the use of ozone as an alternative sanitization method varies among different countries. In USA, ozone has been classified as GRAS (Generally Recognized as Safe) since 1982 for the disinfection treatment of bottled water and in 1995 was used as a sanitizer for process trains in bottled water plants [52]. In 1997, ozone was declared as GRAS for direct contact with foods and in 2001 the FDA approved the use of ozone in gaseous and aqueous form as an antimicrobial for direct contact with different food matrices, including fish, meat, and poultry [52]. In the European Union, instead, there are no specific regulations about the use of the present technology on foods and in processing environments. In Italy, the use of gaseous ozone for the disinfection of cheese ripening rooms has been approved by the Ministry of Health, but the use at direct contact with cheese is not allowed [22]. In Japan, ozone has been used for the treatment of different food plants [55]. Ozone and relative treatments are permitted as processing aids in Australia, New Zealand, Russia, Armenia, Belarus, Kazakhstan, and Kyrgyzstan [8,52].

### 3.6. Limitations: Toxicity, Effect on Foods and Equipment

One of the main issues about the practical application of ozone is related to its toxicity. As ozone is toxic for humans, it is crucial to reduce the exposure of operators to minimize health risks. In this regard, the use of ozone destructors could be convenient when high concentrations are applied. Otherwise, treatments with high ozone concentrations may be applied in the absence of operators or during the weekly closing days [8,23]. In several countries, exposure limits are imposed in order to preserve the operators. In the USA and UK, for example, a continuous exposure limit of 0.1 ppm (8 h/day, 40 h/week) and 0.3 ppm (15 min for a maximum of four times per day) [52] is permitted.

Another potential disadvantage of the ozone use in dairy industries is related to its effect on some food categories. Due to its high oxidative power, ozone could affect the stability of food containing high levels of fat. Sert et al. [56], as example, showed that ozone treatments resulted in a significative antimicrobial effect in butter but also in a decrease in the oxidative stability of the final product. Conversely, Segat et al. [57] showed that ozone treatments did not increase primary and secondary lipid oxidation products in mozzarella cheese.

Another aspect to consider is the effect of ozone on the materials and equipment commonly used in the food industry. High concentrations of ozone, indeed, may cause the corrosion of these materials. In a study [58], the application of pulsed ozone in water at room temperature (20 min for seven days) resulted in weight loss of different materials, including aluminum, carbon steel, copper, and stainless steel (304 and 316), even if this weight loss was significantly greater (a = 0.05) than the control samples only for carbon steel. Conversely, plastics frequently used in food industries, such as PTFE (Teflon), PVDF (Kynar), PVC, and ECTFE (Halar), exhibited resistance to corrosion after exposure to ozone [29]. Basically, treatments with ozone must always be optimized in relation to the materials to be treated, in order to avoid damage to the equipment in the production environment.

### 3.7. Effect on Microbial Biofilms

Few studies have investigated the effect of ozone in the prevention or removal of microbial biofilm. The mechanisms by which the ozone affects bacterial biofilms are still unclear. As highlighted by Moore et al., Gram-negative bacteria are usually more sensitive to ozone than Gram-positive microorganisms [59]. Panebianco et al. [26] hypothesized that a preventive application of ozone on *L. monocytogenes* planktonic cells reduces the capacity of bacteria to produce the extracellular polymeric matrix, while oxidative stress can lead to a reduction of the total biomass in preformed biofilm as a consequence of structural losses of the extracellular matrix. In this section, we report that the main spoilage and pathogenic bacterial groups able to form biofilms in the dairy environment. These main genera are reported also in Table 1. However, we must highlight that, in the dairy environment, mixed biofilms formed by both spoilage and pathogenic bacteria, such as *L. monocytogenes* and *P. fluorescens*, could often be detected [60].

In this section, readers can find relevant data about the action of ozone against the sessile form of bacteria of interest to the dairy industry; data are summarized in Table 2.

#### 3.7.1. *Pseudomonas*

The genus *Pseudomonas* includes bacteria that are able to cause several alterations in milk and dairy products. Specifically, *Pseudomonas* spp. are responsible for undesirable odors and flavors as well as unusual pigments of foods [61]. These microorganisms are ubiquitous; therefore, they are usually isolated at different production stages in the dairy environment. Species more commonly isolated from dairy plants are *P. fluorescens*, *P. koreensis*, *P. marginalis*, *P. rhodesiae*, *P.fragi*, *P. putida*, *P. entomophila*, *P. mendocina*, and *P. aeruginosa* [62]. Even if these bacteria are sensitive to thermal treatments commonly used in dairy processing, thermostable enzymes, such as proteases and lipases, could persist after treatments causing spoilage in finished products [33]. Several studies demonstrated that *Pseudomonas* isolates from milk, dairy products and dairy processing environments are able to form biofilm. In this regard, a recent study highlighted the relationship between biofilm formation abilities and the production of blue pigment of *P. fluorescens* dairy-related strains [63].

Several studies demonstrated how microorganisms belonging to this genus are generally susceptible to ozone exposure even when they are attached to common surfaces or organized in biofilms. As early as 1993, Greene et al. [94] showed that ozonated (0.5 ppm) water treatment (10 min exposure) was effective in reducing the loads (~4 Log10) of common psychrotrophic spoilage bacteria, including *P. fluorescens* and *Alcaligenes faecalis*, on stainless steel surfaces, meanwhile highlighting that the effect of this technology was better performing than the commercial chlorinated sanitizers used in high concentration (100 ppm). Similarly, Dosti et al. [95] reported the effectiveness of ozone treatment (0.6 ppm for 10 min) on *P. fluorescens* (ATCC 948), *P. fragi* (ATCC 4973), *P. putida* (ATCC 795), *Enterobacter aerogenes* (ATCC 35028), *E. cloacae* (ATCC 35030) and *B. licheniformis* (ATCC 14580) on stainless steel coupons. The sensitivity of *P. fluorescens* at ozone treatments was also shown by Marino et al. [8], which demonstrated the effectiveness of ozonated water (0.5 ppm) applied in static as well as dynamic conditions on biofilms. The authors also reported that ozone in gaseous form (20 ppm) led to a reduction of 5.51 Log CFU/cm^2^ after 60 min treatment. Khadre and Yousef [53] studied the effect of ozone on bacterial biofilms and dried films of *B. subtilis* spores and *P. fluorescens* in a multilaminated aseptic food packaging material and stainless steel. Ozone inactivated *P. fluorescens* in biofilms more effectively on stainless steel than on the multilaminated packaging material. Shelobolina et al. [96] studied the effect of dissolved ozone (2, 5 and 7 ppm for 10 and 20 min) on *P. aeruginosa* biofilm grown on glass. The regression equation, used to analyze the effect of ozone, highlighted that biofilm inactivation was correlated to the concentration and the contact time (predicted D-values: 11.1, 5.7 and 2.2 min at 2, 5 and 7 ppm, respectively). The same authors studied the inactivation of biofilms on various surfaces by dissolved ozone (5 ppm for 20 min). The outcomes emphasized that biofilms grown on ceramics were more difficult to inactivate than those grown on plastic materials. Ozone can also be effective on *Pseudomonas* biofilm in combination with other technologies. For example, ozone water in combination with a hydrogen peroxide solution was effective on *P. fluorescens* biofilm. In this regard, a sequential treatment with 1.0 and 1.7 mg/L of ozone followed by 0.8 and 1.1% of hydrogen peroxide showed synergistic disinfection effects [97].

#### 3.7.2. *Bacillus*

Among spore-forming bacteria, the genus *Bacillus* is of high importance, since it includes bacteria that can cause spoilage in milk and dairy products, as well as foodborne pathogens. *Bacillus* species are ubiquitous, Gram-positive, motile, and rod-shaped bacteria, characterized by high versatility and adaptability to different environmental conditions and can survive during the different stages of processing and manufacturing of dairy products [64]. The most common species found in dairy environments are *B. licheniformis*, *B. cereus*, *B. subtilis*, *B. thuringiensis*, *B. weihenstephanensis*, *B. mycoides*, *B. sporothermodurans*, and *B. megaterium* [64,65,66]. *Bacillus* are able to adhere and persist on different surfaces; in addition, they can form other biofilm types, including bundles in the liquid phase and pellicles at the air-liquid interface [67,68]. Additionally, *Bacillus* can form heat-resistant spores that can survive after the routinary pasteurization processes.

Different studies have emphasized how ozone is effective on biofilms formed by dairy-related *Bacillus*. A recent researchdemonstrated the efficacy of gaseous ozone treatment (45 ± 2 ppm) on *B. cereus* biofilms formed on stainless steel and polypropylene [98]. Another study evaluated the effect of ozonated water on *B. cereus* biofilms grown on dairy processing membranes and highlighted an average reduction of 1.0 Log CFU/cm^2^ for treated membranes [99]. The efficacy of ozone, in combination with cleaning in place reagent (NaOH), was shown on biofilms formed by *B. subtilis* and *B. amyloliquefaciens* on stainless steel. Higher inactivation of biofilms (60 and 120″) was obtained with 1.4 ppm of ozone coupled with 1% NaOH as compared to NaOH alone, which required 240 s to completely remove the film from the stainless steel coupons [100].

#### 3.7.3. *Listeria*

This genus comprises one of the most studied bacteria worldwide, that is *L. monocytogenes. L. monocytogenes* is a Gram-positive foodborne pathogen. This pathogen, when organized in established biofilms, can persist over a long period of time on surfaces and food processing environments, thus representing a potential cause of repeated contaminations of the finished products [74,75,76]. Detection of biofilm forming and persistent *L. monocytogenes* strains in the dairy environment was reported in several studies [45,46]. It has been demonstrated that adhesion capacity and biofilm formation abilities differ among several *L. monocytogenes* strains. This strain variability seems to be linked with the presence of specific genes and/or accessory genetic elements, such as phages, plasmids and stress survival islets [26,77].

Several experiments have been performed so far on *L. monocytogenes* biofilm, highlighting that high ozone levels and long exposure times are needed to achieve an effect against biofilms formed by this bacterium. Korany et al. [101] reported that ozonated water treatment (1 min at 1.0, 2.0 and 4.0 ppm) resulted in ∼0.9, 3.4 and 4.1 log reduction of *L. monocytogenes* single strain biofilm on polystyrene, but the effect was lower with multi-strain biofilms and in the presence of organic matter. Robbins et al. [102] obtained a complete elimination of attached *L. monocytogenes* Scott A and 10403S strains cells after exposure to 4 ppm of ozone. Nicholas et al. [103] reported a mean reduction of 3.41 Log10 CFU/cm^2^ for stainless steel-attached *L. monocytogenes* cells after 1 h treatment at 45 ppm of gaseous ozone, but the same strains organized in biofilm were significantly more resistant after a treatment with ozone gas at 45 ppm for 1 h. Harada et al. [104] demonstrated the efficiency of gaseous ozone (45 ppm) as a dry sanitizing method on *L. monocytogenes*. The authors observed a reduction of sessile cells below the limit of detection (1.7 Log CFU/cm^2^) in 5 min on polypropylene, while a reduction of 3.4 Log CFU/cm^2^ was observed in stainless steel. A recent experiment conducted on dairy- and meat-related *L. monocytogenes* showed that ozone gas in high concentrations (50 ppm for 6 h) caused a significant decrease of the biofilm biomass for 59% of the strains tested, but only a slight reduction of live cells in the formed biofilm was observed [26]. De Candia et al. [21] demonstrated the efficacy of cold gaseous ozone treatments at low concentrations in the eradication of *L. monocytogenes* from different food contact surfaces (glass, polypropylene, stain-less steel, expanded polystyrene). A continuous ozone flow (1.07 mg m^−3^) after 24 or 48 h of cold incubation resulted in the inactivation of 11 strains, while with higher inoculum levels (9 log CFU coupon^−1^) the best inactivation rate was detected after 48 h of treatment at 3.21 mg m^−3^ of ozone on stainless steel and expanded polystyrene. Baumann et al. [105] tested ozone (concentrations of 0.25, 0.5 and 1.0 ppm) in combination with power ultrasound cycled through 250 mL of a potassium phosphate buffer containing *L. monocytogenes* biofilm chips for 30 or 60 s. Reductions obtained with the combined treatments were significantly (*p* < 0.05) higher than each treatment alone. No recoverable cells were detected (reduction = 7.31 Log CFU/mL) after 60 s of the combined treatment when ozone was used at a concentration of 0.5 ppm.

#### 3.7.4. *Staphylococcus*

*S. aureus* is deemed as an important pathogen detectable in dairy products. Several studies showed how this bacterium can form biofilm in the dairy environment. Lee et al. [78] studied the biofilm production abilities of strains isolated from milking parlor environments on dairy farms in Brazil. Around 45% of *S. aureus* pulsotypes were able to form biofilms in at least one assay, suggesting their possible persistence in milking environments. Biofilm forming abilities were demonstrated also for dairy-related *S. aureus* isolated from Switzerland and Italy, including methicillin-resistant *S. aureus* (MRSA) [79], and for strains isolated from food-contact surfaces in dairy industries in Mexico [80].

*S. aureus* is generally sensitive to ozone exposure. Cabo et al. demonstrated that the application of 1 μg/g of ozonized water allowed 99% inactivation in 2 min of *S. aureus* CECT4459 biofilm on polypropylene [106]. Shao et al. [107] studied the effect of ozone water on mature *S. aureus* and *Salmonella* spp. biofilm, detecting less than 0.8 Log cfu/cm^2^ of cells reduction in biofilm exposed to ozonized water for 20 min. In the study performed by Marino et al. [8], *S. aureus* was highly responsive to aqueous ozone treatments at dynamic conditions, while exposure to gaseous ozone at high concentrations (20 ppm) resulted in a reduction of 4.72 Log CFU/cm^2^ of *S. aureus* biofilm. A recent study investigated the effect of ozonated oils with concentrations ranging from 0.53 to 17 mg/g on Methicillin-resistant *S. aureus* (MRSA) biofilm; most strains were inhibited at concentrations of 4.24 mg/g. Additionally, ozonated oils showed ability in removing adherent cells and high capacity in the eradication of 24 h biofilms [108].

#### 3.7.5. *Salmonella*

The genus *Salmonella* includes well known pathogenic bacteria which can be found in different types of foods. Historically, several *Salmonella* outbreaks were linked to the consumption of dairy products, especially raw milk products [81,82,83]. Bacteria of this genus, indeed, can persist in fresh and fermented dairy products for their adaptation to an acid environment. Leyer and Johnson [84] demonstrated that acid-adapted *S. typhimurium* cells had increased resistance to organic acids usually present in cheese, such as lactic, propionic, and acetic acid. In a study of Kessel et al. [85], *Salmonella* was isolated from 36 of 75 PCR-positive bulk tank milk samples and 105 of 174 PCR-positive milk filter samples. Lamas et al. [86] proved that milk residues are a source of nutrients for *S. enterica* biofilm formation on stainless steel and that the biofilm forming abilities of this bacterium are strongly related to oxygen levels.

Few studies exist about the effectiveness of ozone treatments against biofilms formed by bacteria belonging to this genus. Shao et al. [107] reported a reduction less than 0.8 Log cfu/cm^2^ of *S. aureus* and *Salmonella* spp. biofilm after exposure to ozonized water for 20 min. In another study [109], the effect of malic acid and ozone against *S. typhimurium* biofilm on different food contact surfaces (PVC pipes, polyethylene, plastic, and fresh produce) was explored. The mutual effect of malic acid with ozone resulted in a reduction of biofilm formation on plastic bags and PVC pipes. In microtiter plates, reductions in biofilm formation were observed after 20 h and 40 h treatments.

#### 3.7.6. *Clostridium*, *Cronobacter*, *Escherichia*

The genus *Clostridium* includes Gram-positive spore-forming anaerobes bacteria that can induce spoilage of dairy products by gas production arising from the fermentation of acetate, lactate, and butyrate [69,70]. *C. tyrobutyricum* is considered the species most frequently involved in cheese spoilage, as it is the causative agent of the so called “late blowing defect”, though *C. sporogenes*, *C. beijerinckii*, *C. tyrobutyricum*, and *C. butyricum* can also cause cheese alterations [69,70]. Additionally, this genus includes pathogenic bacteria, such as *C. botulinum* and *C. perfringens*. A recent study showed that dairy-related *C. perfringens* isolates were able to form biofilm at different temperatures (4, 25, and 35 °C) [71]. The genus *Cronobacter* comprises *C. sakazakii*, a relevant foodborne pathogen included in the food safety criteria for infant foods in the Regulation EC 2073/05 and amendments [110]. This bacterium is characterized by high adaptability to the dairy environment. This aspect was emphasized by Oh et al. [72], which studied the biofilm-forming abilities of 72 strains on plastic surfaces, as well as the influence of the artificial growth medium and infant milk formula (IMF). The diversity and biofilm forming abilities of *Cronobacter* isolated in New Zealand were investigated by Gupta et al. [73], which showed that adherence characteristics are related with nutrients and temperature. The genus *Escherichia* includes relevant foodborne pathogens associated with dairy products, such as the enterohemorrhagic *E. coli* O157:H7. Several studies demonstrated that this bacterium could survive in different types of dairy products, and *E. coli* O157:H7 outbreaks frequently occurred after consumption of unpasteurized cheese [87,88,89,90]. Sharma and Anand analyzed biofilms of pasteurization lines in a commercial plant and in an experimental dairy plant, revealing the presence of *E. coli* in both cases [91]. The high biofilm forming abilities of different Shiga toxin–producing *E. coli* (STEC) and the strong tolerance to common sanitizers led to concerns regarding the colonization of surfaces and the resultant downstream food contamination [92,93].

To the best of our knowledge, no relevant studies describing the effect of gaseous ozone against biofilms of these bacteria have been conducted, while the effect of ozone on the vegetative and spore forms is well documented. Foegeding [111] studied the effect of ozone on spores of *C. perfringens* NCTC 8798 and *C. botulinum* 12885A strains, highlighting how ozone was an effective sporicide, especially at acidic pH values. The action of ozone improved the efficiency of cooking temperatures (from 45 to 75 °C) against *C. perfringens* on beef surfaces [112]. Significant reductions of vegetative cells (from 5.59 ± 0.17 to 4.09 ± 0.72 and 3.50 ± 0.90 log CFU/g after treatments with aqueous ozone at 5 ppm and heating at 45 and 55 °C) were reported. Spores, indeed, were reduced from 2.94 ± 0.37 log spores/g to 2.07 ± 0.38 log spores/g and 1.70 ± 0.37 log spores/g after the treatments with 5 ppm of aqueous ozone and heating at 55 and 75 °C, respectively. Gaseous ozone was effective in the inactivation of *Cronobacter* in milk powders. A continuous stream of ozone led to a reduction of 2.71 and 3.28 log after 120 min at 2.8 and 5.3 mg/L^−1^, respectively [113]. With regards to *Escherichia*, a study carried out in 2010 did not reveal any significative effect of ozone (2 mg/L) in removing *E. coli* and *L. monocytogenes* cells in biofilms on lettuce surfaces [114]. However, in the study of de Oliveira Souza et al. [115], ozonated water (35 and 45 mg/L^−1^ for 0, 5, 15, and 25 min) was effective in inactivating *E. coli* O157:H7, while reductions of 1.5 log cycles were detected in lactose-free homogenized skim milk, indicating the influence of the substrate on the antimicrobial efficiency of ozone. Effectiveness of aqueous ozone treatment (5 mg/L) on Shiga toxin-producing *E. coli* inoculated in alfalfa seeds was demonstrated by Mohammad et al. [116], who reported mean log reductions of 1.5 ± 0.4, 1.6 ± 0.4, 2.1 ± 0.5 after 10, 15, and 20 min, respectively.

## 4. Conclusions

The maintenance of good hygienic conditions in the working environment is crucial to avoid microbial contamination by spoilage or pathogenic bacteria of dairy products. Although some reviews have been published on the use of ozone in controlling microbial contamination in the food industry, to our knowledge, this is the first review focused on the effect of ozone against microbial biofilms, commonly found in dairy plants. Currently, data concerning the efficacy of ozone as an anti-biofilm agent in the food context suggest that the anti-biofilm action is variable and still not completely investigated. The advantages of using ozone in dairy processing plants include the low environmental impact, which comprise the absence of harmful residuals in food products or on contact surfaces. Nevertheless, further studies are needed to evaluate its action in preventing or removing microbial biofilm both in experimental and under realistic environmental conditions. Finally, since ozone application has several limitations (toxicity, potential effect on materials and high-fat foods), restrictions and detailed application protocols should be applied in relation to the specific needs and characteristics of dairy processing environments.

## Figures and Tables

**Table 1 microorganisms-10-00162-t001:** Bacteria able to form biofilms in the dairy environment.

Genus	Species Commonly Found in Dairy Products and Environment	Problems	References
*Pseudomonas*	*P. fluorescens, P. koreensis, P. marginalis, P. rhodesiae, P. fragi, P. putida, P. entomophila, P. mendocina, P. aeruginosa*	Spoilage: *P. fluorescens, P. koreensis, P. marginalis,* *P. rhodesiae, P. fragi, P. putida, P. entomophila,* *P. mendocina* Foodborne pathogens: *P. aeruginosa*	[33,61,62,63]
*Bacillus*	*B. licheniformis, B. cereus, B. subtilis, B. thuringiensis, B. weihenstephanensis, B. mycoides, B. sporothermodurans, B. megaterium*	Spoilage: *B. licheniformis, B. cereus, B. subtilis,* *B. thuringiensis, B. weihenstephanensis, B. mycoides,* *B. sporothermodurans, B. megaterium* Foodborne pathogens: *B. cereus*	[64,65,66,67,68]
*Clostridium*	*C. tyrobutyricum, C. sporogenes, C. beijerinckii, C. butyricum, C. botulinum, C. perfringens*	Spoilage: *C. tyrobutyricum, C. sporogenes,* *C. beijerinckii, C. butyricum* Foodborne pathogens: *C. botulinum, C. perfringens*	[69,70,71]
*Cronobacter*	*C. sakazakii*	Foodborne pathogen	[72,73]
*Listeria*	*L. monocytogenes*	Foodborne pathogen	[26,45,46,74,75,76,77]
*Staphylococcus*	*S. aureus*	Foodborne pathogen	[78,79,80]
*Salmonella*	*S. typhimurium, S. enterica*	Foodborne pathogens	[81,82,83,84,85,86]
*Escherichia*	Shiga toxin–producing *E. coli* (STEC)	Foodborne pathogens	[87,88,89,90,91,92,93]

**Table 2 microorganisms-10-00162-t002:** Data on the effect of ozone on biofilms formed by dairy-related spoilage and pathogenic bacteria.

Target Microorganisms		Surface/Material	Treatment	Effect	Reference
Genus	Species				
*Pseudomonas*	*P. fluorescens*	Stainless steel	Ozonated water (0.5 ppm for 10 min)	Loads Reduction (~4 Log10)	[94]
	*P. fluorescens, P. fragi, P. putida*	Stainless steel	Ozonated medium (0.6 ppm for 10 min)	Loads Reduction (from 2.9 to 4.2 Log CFU/cm^2^)	[95]
	*P. fluorescens*	Stainless steel	(i) Static: ozonated water (0.5 mg/L) at 20 s, 40 s, 1 min, 3 min, 5 min, 10 min, and 20 min. (ii) Dynamic: flow of ozonated water (0.5 mg/L) for 20 s, 40 s, 1 min, 3 min, 5 min, 10 min, and 20 min. (iii) Gaseous ozone: concentrations of 0.1, 0.15, 0.2, 2, 5, and 20 ppm for exposure times of 2, 5, 7, 10, 20, 30, and 60 min.	(i) Loads Reduction (~1.56 Log CFU/cm^2^ in 20 min); (ii) Loads Reduction (~3.52 Log CFU/cm^2^ in 20 min); (iii) Loads Reduction (~5.51 Log CFU/cm^2^ in 20 min)	[8]
	*P. fluorescens*	Multilaminated food packaging, stainless steel	Aqueous ozone (3.7–12.9 mg/mL)	Decrease of 2.3–2.6 logs after 1 min exposure to 4.5–5.6 mg/mL. More efficiency on stainless steel compared to the multilaminated packaging material (difference of 2–4 logs depending on the dosage of ozone)	[53]
	*P. aeruginosa*	Glass, ceramic, plastic	Dissolved ozone (2, 5 and 7 ppm for 10 and 20 min)	Inactivation correlated to the concentration and the time (predicted D-values: 11.1, 5.7 and 2.2 min at 2, 5 and 7 ppm, respectively). Inactivation (5 ppm for 20 min)	[96]
	*P. fluorescens*	Glass	Sequential treatment (1.0 and 1.7 mg/L of ozone followed by 0.8 and 1.1% of hydrogen peroxide)	Significative effect on the survival ratio	[97]
*Bacillus*	*B. cereus*	Stainless steel, polypropylene	Gaseous ozone (45 ± 2 ppm for 30 min)	Greater action on stainless steel in the first 10 min. Polypropylene: increase in the reduction with the exposure time, until 2.16 Log CFU/cm^2^ after 30 min	[98]
	*B. cereus*	Flat sheet polyethersulfone (PES) membranes	Ozonated water	Average reduction of 1.0 Log CFU/cm^2^	[99]
	*B. subtilis,* *B. amyloliquefaciens*	Stainless steel	Gaseous ozone (1.4 ppm) in combination with cleaning in place reagent (NaOH)	Higher inactivation (60 and 120″) obtained with 1.4 ppm of ozone coupled with 1% NaOH as compared to NaOH (1%) alone (240″)	[100]
*Listeria*	*L. monocytogenes*	Polystyrene	Ozonated water (1.0, 2.0, and 4.0 ppm for 1 min)	∼0.9, 3.4, and 4.1 Log reduction	[101]
	*L. monocytogenes*	Stainless steel	Ozonated PPB (3 min)	Attached cells eliminated at concentration of 4.00 ppm (7.47-log reduction). A fourfold increase in sanitizer concentration was required to destroy biofilm cells	[102]
	*L. monocytogenes*	Stainless steel	Gaseous ozone (45 ppm)	Mean reduction of 3.41 Log10 CFU/cm^2^ for stainless steel-attached cells after 1 h. The same strains organized in biofilm were significantly more resistant	[103]
	*L. monocytogenes*	Polypropylene, stainless steel	Gaseous ozone (45 ppm)	Reduction of sessile cells below the limit of detection (1.7 Log CFU/cm^2^) in 5 min on polypropylene; reduction of 3.4 Log CFU/cm^2^ in stainless steel	[104]
	*L. monocytogenes*	Polystyrene	Gaseous ozone (50 ppm for 6 h)	Significant decrease of the biofilm biomass (colorimetric assay) for 59% of the strains tested; slight reduction of live cells in the formed biofilm	[26]
	*L. monocytogenes*	Glass, polypropylene, stainless steel, expanded polystyrene	Cold gaseous ozone	A continuous ozone flow (1.07 mg m^−3^) after 24 or 48 h of cold incubation resulted in the inactivation of 11 strains; with high inoculum level (9 log CFU coupon^−1^) the best inactivation rate was observed after 48 h of treatment at 3.21 mg m^−3^ of ozone on stainless steel and expanded polystyrene	[21]
	*L. monocytogenes*	Stainless steel	Ozone in combination with power ultrasound treatment	Reductions of combined treatments were significantly (*p* < 0.05) higher than by either treatment alone. No recoverable cells after 60 s of combined treatment when an ozone concentration of 0.5 ppm was used (7.31-log CFU/mL reduction)	[105]
*Staphylococcus*	*S. aureus*	Polypropylene	Ozonized water (1 mg/g)	99% inactivation	[106]
	*S. aureus*	Stainless steel	Ozonized water	Reduction less than 0.8 Log CFU/cm2 of *S. aureus* and *Salmonella* spp. biofilm after exposure to ozonized water for 20 min	[107]
	*S. aureus*	Stainless steel	(i) Static: ozonated water (0.5 mg/L) at 20 s, 40 s, 1 min, 3 min, 5 min, 10 min, and 20 min. (ii) Dynamic: flow of ozonated water (0.5 mg/L) for 20 s, 40 s, 1 min, 3 min, 5 min, 10 min, and 20 min. (iii) Gaseous ozone: concentrations of 0.1, 0.15, 0.2, 2, 5, and 20 ppm for exposure times of 2, 5, 7, 10, 20, 30, and 60 min.	Highly sensitive to aqueous ozone treatment at dynamic conditions; exposure to gaseous ozone at high concentrations (20 ppm) resulted in a reduction of 4.72 Log CFU/cm^2^ of biofilm	[8]
	Methicillin-resistant *S. aureus* (MRSA)	Polystyrene	Ozonated oils (from 0.53 to 17 mg/g)	Most strains inhibited at concentrations of 4.24 mg/g. Removal of adherent cells and high capacity in the eradication of 24 h biofilms	[108]
*Salmonella*	*S. Agona, S. Infantis, S. Typhimurium, ATCC 13076 (Enteritidis serotype), S. Enteritidis*	Stainless steel	Ozonized water	Reduction less than 0.8 Log CFU/cm^2^ of *S. aureus* and *Salmonella* spp. biofilm after exposure to ozonized water for 20 min	[107]
	*S. typhimurium*	PVC pipes, polyethylene, plastic, fresh produce	Malic acid and ozone	Reduction of biofilm formation on plastic bags and PVC pipes. In microtiter plates, reductions in biofilm formation were observed after 20 h and 40 h treatments	[109]

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
