# Peer review of "The Use of Ozone as an Eco-Friendly Strategy against Microbial Biofilm in Dairy Manufacturing Plants: A Review"

_microorganisms, 2022, doi:10.3390/microorganisms10010162_

Round 1

Reviewer 1 Report

This is a very interesting review on the role of ozone for microbial growth control in the food sector.

However, for a review manuscript, an increase in the number of references is recommended. The role of biofilm in the dairy sector could be further dissecated. It is further recommended to describe mechanistic information on the action of ozone against biofilms. The manuscript would benefit from the inclusion of a Figure demonstrating such an action.

Minor language editing is recommended.

Author Response

Dear Editor and reviewers,

We would like to thank the reviewers for their interesting suggestions regarding our manuscript. In the revised version of the manuscript most of these suggestions have been addressed. We revised the manuscript in line with the reviewer’s comments. Below our response on the changes made within the manuscript. We tried to answer to all the reviewer's questions. Our answers follow the reviewer’s comments and are highlighted in red.

Reviewer 1

This is a very interesting review on the role of ozone for microbial growth control in the food sector.

However, for a review manuscript, an increase in the number of references is recommended.

Thank you for the suggestion. We added other authors in the reference list.

The role of biofilm in the dairy sector could be further dissecated.

Thank you for the suggestion. The role of biofilm in the dairy sector has been further explored in the new section 2.Biofilm occurrence in the dairy industry”. More information on this topic were added in the manuscript.

It is further recommended to describe mechanistic information on the action of ozone against biofilms. The manuscript would benefit from the inclusion of a Figure demonstrating such an action.

Thank you for the suggestion. To the best of our knowledge, in the literature there are very few information about the specific action of ozone against microbial biofilm. However, we reported some hypothesis about the possible mechanism of action of ozone against biofilm at the beginning of the paragraph 3.4.7. “Effect on microbial biofilms”.

Minor language editing is recommended.

Thank you for the suggestion. The language was thoroughly revised in several parts of the manuscript.

Reviewer 2 Report

This review shows a thorough description of research published on the use of ozone as antimicrobial, including research previously conducted by the authors. The two major issues that need to be addressed in a revision of this manuscript are:

  1. The review seems to be somewhat misleading by proposing ozone as an ecologically beneficial alternative to other chemical sanitizers, particularly regarding the potential effect on cross-resistance to other antimicrobials including antibiotics. While this is a highly relevant issue, the literate cited on the lower risk for developing antimicrobial resistance when using ozone was not sufficiently substantiated. The cite: "Ozone, in fact, is less amenable to induce antimicrobial resistance compared to conventional chemical agents" did not present any reference to substantiate this claim. Authors should add more text to address what is stated on the title (eco-friendly alternative). If this claim can't be supported, authors may consider changing the title and premise of the review.
  2. The organization of the manuscript seemed to make it difficult to follow by the reader. There are two subsections where individual pathogens are described, making reading somewhat tiresome. The authors may consider reorganizing, merging the description of individual pathogens in a single subsection that addresses their ability to form biofilms, as well as  studies on the effectiveness of ozone against these pathogens.
  3. A third issue, which this reviewer considered to be minor because is easy to resolve (although highly important), is the rather less-than-perfect stage of English in the manuscript. Although i general the text can be followed, several grammatical errors render the manuscript unacceptable. Authors are encouraged to have the revised manuscript proof-read by someone with proven English skills before submitting to Microorganisms

Specific, details to correct are:

Lines 64-65. Provide reference(s) to support this claim.

Lines 335-338. Rewrite this sentence to clarify intended message. The relation between corrosion-resistant plastics and other materials potentially susceptible to corrosion in unclear.

Author Response

Dear Editor and reviewers,

We would like to thank the reviewers for their interesting suggestions regarding our manuscript. In the revised version of the manuscript most of these suggestions have been addressed. We revised the manuscript in line with the reviewer’s comments. Below our response on the changes made within the manuscript. We tried to answer to all the reviewer's questions. Our answers follow the reviewer’s comments and are highlighted in red.

Reviewer 2

This review shows a thorough description of research published on the use of ozone as antimicrobial, including research previously conducted by the authors. The two major issues that need to be addressed in a revision of this manuscript are:

The review seems to be somewhat misleading by proposing ozone as an ecologically beneficial alternative to other chemical sanitizers, particularly regarding the potential effect on cross-resistance to other antimicrobials including antibiotics. While this is a highly relevant issue, the literate cited on the lower risk for developing antimicrobial resistance when using ozone was not sufficiently substantiated. The cite: "Ozone, in fact, is less amenable to induce antimicrobial resistance compared to conventional chemical agents" did not present any reference to substantiate this claim. Authors should add more text to address what is stated on the title (eco-friendly alternative). If this claim can't be supported, authors may consider changing the title and premise of the review.

Thank you for your precious suggestion. In the manuscript some sentences were not clearly written. To avoid misunderstandings that can generate confusion, we modified this part in the text.

Ozone is considered as an alternative technology with a low environmental impact, we have included a sentence and references to support this statement (L68-72 in the revised version of the manuscript). However, we agree with reviewer’s comment. The topic of antimicrobial resistance should be further investigated. It is not the main focus of our review. In brief, we have changed the title and removed the sentence: “Ozone, in fact, is less amenable to induce antimicrobial resistance compared to conventional chemical agents” in the text.

The organization of the manuscript seemed to make it difficult to follow by the reader. There are two subsections where individual pathogens are described, making reading somewhat tiresome. The authors may consider reorganizing, merging the description of individual pathogens in a single subsection that addresses their ability to form biofilms, as well as  studies on the effectiveness of ozone against these pathogens.

Thank you for the suggestion. The manuscript was reorganized merging the descriptions of biofilm-forming spoilage and pathogenic bacteria and the effect of ozone against these microorganisms in a single section (please see 3.4.7. Effect on microbial biofilms in the revised version of the manuscript).

A third issue, which this reviewer considered to be minor because is easy to resolve (although highly important), is the rather less-than-perfect stage of English in the manuscript. Although i general the text can be followed, several grammatical errors render the manuscript unacceptable. Authors are encouraged to have the revised manuscript proof-read by someone with proven English skills before submitting to Microorganisms.

Thank you for the suggestion. The language was thoroughly revised in several parts of the manuscript.

Specific, details to correct are:

Lines 64-65. Provide reference(s) to support this claim.

Thank you for the suggestion. This sentence was removed.

Lines 335-338. Rewrite this sentence to clarify intended message. The relation between corrosion-resistant plastics and other materials potentially susceptible to corrosion in unclear.

Thank you for the suggestion. The sentence was rephrased clarifying the message and including other relevant references (L278-287 in the revised version of the manuscript).